# Baby Foods: 9 Out of 62 Exceed the Reference Limits for Acrylamide

**DOI:** 10.3390/foods13172690

**Published:** 2024-08-26

**Authors:** Arianna Bonucci, Stefania Urbani, Maurizio Servili, Roberto Selvaggini, Luigi Daidone, Ilenia Dottori, Beatrice Sordini, Gianluca Veneziani, Agnese Taticchi, Sonia Esposto

**Affiliations:** Department of Agriculture, Food and Environmental Sciences, University of Perugia, 06126 Perugia, Italy; arianna.bonucci@unito.it (A.B.); stefania.urbani99@gmail.com (S.U.); roberto.selvaggini@unipg.it (R.S.); luigi.daidone@unipg.it (L.D.); ilenia.dottori@unipg.it (I.D.); beatrice.sordini@unipg.it (B.S.); gianluca.veneziani@unipg.it (G.V.); agnese.taticchi@unipg.it (A.T.); sonia.esposto@unipg.it (S.E.)

**Keywords:** acrylamide, baby food, estimated daily intake, Maillard reaction, probable carcinogen, neurotoxicity

## Abstract

Acrylamide (AA) is a contaminant resulting from the Maillard reaction and classified by the International Agency for Research on Cancer (IARC) as a probable carcinogen in Group 2A, with proven neurotoxic effects on humans. European Union (EU) Regulation No. 2017/2158 is currently in force, which establishes measures meant to reduce AA levels in food and sets reference values, but not legal limits, equal to 40 and 150 μg/kg AA in processed cereal-based foods intended for infants and young children and in biscuits and rusks, respectively. For this reason, sixty-two baby foods were analyzed using ultra-high performance liquid chromatography with diode array detector and quadrupole time-of-flight mass spectrometry (UHPLC-DAD-Q-TOF/MS) to check whether industries were complying with these values, even though AA control is not legally mandatory. In total, 14.5% of the samples exceeded the reference values; these were homogenized chicken products (211.84 ± 16.53, 154.32 ± 12.71, 194.88 ± 7.40 μg/kg), three biscuits (276.36 ± 0.03, 242.06 ± 0.78, 234.78 ± 4.53 μg/kg), a wheat semolina (46.07 ± 0.23 μg/kg), a homogenized product with plaice and potatoes (45.52 ± 0.28 μg/kg), and a children’s snack with milk and cocoa (40.95 ± 0.32 μg/kg). Subsequently, the daily intake of AA was estimated, considering the worst-case scenario, as provided by the consumption of homogenized chicken products and biscuits. The results are associated with margins of exposure (*MOE*s) that are not concerning for neurotoxic effects but are alarming for the probable carcinogenic effects of AA.

## 1. Introduction

The food industry produces a wide variety of baby foods; however, considering that the consumer is a vulnerable individual with an unstable metabolism and a developing digestive and endocrine system, precautionary principles must be strictly respected. Regulations set stricter standards for toxicological purposes; lower limits are imposed for contaminants in baby food, as reported in European Union (EU) Regulation No. 2023/915, such as heavy metals, pesticide residues, nitrates, dioxins, mycotoxins, and others [1]. For acrylamide (AA), on the other hand, there are no legal limits; the relevant legislation provides only reference values of 150 μg/kg in biscuits and rusks intended for infants and young children and 40 μg/kg in processed cereal-based baby foods, as listed in Annex IV of EU Regulation No. 2017/2158. These values are used to assess the effectiveness of the control measures taken by the industry, and if they are exceeded, the competent authority must carry out a specific risk assessment without any legal impact on the producing industry [2].

Once ingested, AA is absorbed by the gastrointestinal system, distributed, and metabolized in various vital organs [3]. For the most part, AA undergoes a conjugation reaction with glutathione, which loses its antioxidant properties, negatively affecting the redox state of cells [4]. The metabolite requiring most the attention is glycidamide, which results from the oxidation of AA by the enzyme cytochrome P450 oxidase. This metabolite is believed to be the most likely cause of genetic mutations and tumors demonstrated in animals exposed to AA. Both AA and glycidamide can bind to hemoglobin and interact with DNA, causing possible neoplasms in animals [5]. AA remains in the blood for two hours [6], after which small amounts are excreted in the bile and feces, and approximately 60% in the urine within one day after intake. Of this, 90% of AA is excreted as conjugated mercapturic acid metabolites and less than 2% as unmodified AA [7].

Repeated-dose toxicity studies of AA in animals have reported the adverse effects of neurotoxicity and carcinogenicity, the latter of which also occurs in the presence of glycidamide, which confirms that it is the proximal carcinogenic metabolite of AA. The only confirmed adverse effect that has been shown in human studies is neurotoxicity [8], while results are confusing regarding its carcinogenic effects. Only a few studies have suggested an increased risk of prostate, bladder, and kidney cancers in AA-exposed individuals [9]. Others have found adverse effects in renal cells, pre- and postnatal development and an increased risk for endometrial and ovarian cancer, but the evidence are limited and inconsistent. In June 2015, the CONTAM Panel (Panel on Contaminants in the Food Chain) of the European Food Safety Authority (EFSA) published the first comprehensive risk assessment of AA in food, identifying four possible adverse effects, including toxicity (i.e., neurotoxicity, also confirmed in humans), male reproductive effects, developmental toxicity, and carcinogenicity, for which further studies are needed [10]. Given the conflicting results of studies in humans, the International Agency for Research on Cancer has classified AA in Group 2A, listing it as a “probable human carcinogen” [11].

The literature reports many studies investigating the AA content in high-temperature processed foods intended for adults, such as breads, biscuits and other baked goods, potato chips, and coffee [12,13]. Fewer are those that focus on baby food, which are part of the added dietetic or atypical products (ADAP) category and needs to be monitored more closely [14].

This work assessed the concentration of AA in foods intended for infants (subjects younger than 12 months) and young children (aged between 1 and 3 years): infant and follow-on formulas; cereal-based foods; and other foods intended for infants and young children, including growth milk. It was investigated whether values of 150 μg/kg AA in biscuits and rusks and 40 μg/kg AA in other children’s foods, including cereal-based ones, are respected. Then, variations in AA concentrations were assessed based on ingredients, production, and food storage techniques. Finally, the daily intake of AA from consumption of the foods that showed the highest concentration in the study was estimated to assess the worst-case scenario using risk characterization.

## 2. Materials and Methods

### 2.1. Products

Sixty-two baby foods were analyzed (Appendix A), including nine homogenized fruit foods (OFR), five homogenized vegetable foods (OVE), two homogenized cheese foods (OFO), ten homogenized meat foods (chicken-OCP, ham-OCPR, veal-OCV), four homogenized fish foods (OPE); four fruit purees (FRU), two semolina (SE), two creamed rice (CR), one vegetable broth (BV), six biscuits (B), three snacks (SN), four infant and follow-on powder formulas (LP), four growth milk (LC), two milk and cocoa snacks (ME), and four freeze-dried homogenized meat products (LIO). They were purchased in supermarkets and pharmacies in Umbria (Italy). All solid samples were crushed before analysis using an electric grinder MQ30 (Braun, Kronberg im Taunus (DE)). For each sample, two replicates of the AA extraction were performed and analyzed individually using LC-MS.

### 2.2. Reagents and Standards

The following solvents were used to perform the analyses: acetonitrile-grade high-performance liquid chromatography (HPLC) and liquid chromatography–mass spectrometry (LC/MS)-grade water was purchased from Carlo Erba (Carlo Erba Reagents, Cornaredo (MI)); laboratory-grade water was obtained from an Elga Purelab Option R15 purifier (UK); formic acid for LC/MS (used as an additive), acrylamide (purity ≥ 99%), and methacrylamide (purity ≥ 99%) were purchased from Merck Life Science (Milan, Italy).

### 2.3. Extraction and Evaluation of Acrylamide from Various Food Matrices

AA was extracted from the different matrices as reported by Zhao, 2019 [13] with the following modifications: 100 μL of an internal standard solution consisting of methacrylamide (100 mg/L) was added to the samples (1 g) to achieve a final concentration of 500 µg/mL. The sample was then mixed with 10 mL of distilled water and 10 mL of acetonitrile (in the case of liquid milk, 10 mL was taken without adding water) and vortexed 10 min at 2500 rpm. Then, 1.5 g of a mixture of salts consisting of MgSO_4_ plus NaCl 4:1 (*p*:*p*) was added, after which, the solution was vortexed at 2500 rpm for 2 min. Then, the sample was centrifuged at 4000 rpm for 6 min, and the separated supernatant was taken and stored at −20 °C until analysis. Quantitative analysis of AA present in various food matrices was performed with an ultra-high performance liquid chromatography with diode array detector and quadrupole time-of-flight mass spectrometry (UHPLC-DAD-Q-TOF/MS). The Agilent Technologies model 1260 Infinity was used and includes a degasser, binary pump, autosampler, column thermostatic oven, and diode array detector (DAD), all coupled to an Agilent 6530 Accurate-Mass Q-TOF LC/MS model quadrupole-time-of-flight (Q-TOF) mass spectrometer with a dual jet stream electrospray ionization (ESI) ionization source (Agilent Technologies, Santa Clara, CA, USA). The column was a Zorbax Eclipse Plus C18 100 mm × 2.1 mm, 1.8 μm (Agilent Technologies, Santa Clara, CA, USA). The sample extracts were diluted (1:1 *v*/*v*) and filtered through 0.2 μm polyvinylidene fluoride (PVDF) syringe filters with a diameter of 25 mm (Carlo Erba Reagents, Cornaredo, MI, IT). The volume of the sample injected was 5 μL, and elution was performed at a flow rate of 0.3 mL/min using water-fortified 0.1% formic acid as solvent A and methanol with 0.1% formic acid as solvent B. The elution gradient varied as follows: 0 min, 95% phase A and 5% phase B, held for 5 min; switched to 0% A and 100% B over 3 min. It was then returned to the initial conditions, and the system was allowed to equilibrate again for 7 min. The analysis time was 25 min, and the acquisition time was 18 min. The mass spectrum was acquired using ESI ionization in positive mode in an *m*/*z* range of 40–1600 with a scan rate of 1.2 spectra/s, infusing both the eluent from the HPLC system (via the first nebulizer) and the reference mixture (via the second nebulizer) with two masses of *m*/*z* 121.050873 and 922.009798. The dual jet stream ESI source parameters were as follows: sheath gas temperature, 300 °C; sheath gas flow, 12 L/min; dry gas temperature, 250 °C; drying gas flow, 12 L/min; nebulizer pressure, 35 psig; capillary voltage (VCap), 4000 V; nozzle 0 V; fragmentor, 110 V; skimmer, 65 V; and octapole 1 RF, 750 V. The data were acquired in MS/MS mode using a quadrupole to select an acrylamide ion with *m*/*z* 72.0044 and an methacrylamide ion with *m*/*z* 86.06 (precursor ions). applying collision energy values of 10 V and 12 V, respectively. Product ions were extracted with *m*/*z* 55.0192 for acrylamide and *m*/*z* 58.0661 (product ions). Agilent MassHunter B. 10.00 software was used to perform the analysis and to identify and quantify the compounds. The quantitation of acrylamide was made by constructing an eight-points calibration curve using the internal standard method with concentrations of 2.3, 4.7, 9.4, 18.8, 37.5, 75.0, 150.0, and 300.0 µg/L. The limits of detection (LOD) and quantification (LOQ) were calculated separately using the standard deviation (σ) of the replicated responses at the lowest concentration and the slope of the calibration curve (S) using the formula: LOD = 3 · σ/S. The LOQ was calculated as 10 · σ/S. LOD and LOQ were 0.60 and 2.01 µg/kg in the samples, respectively. The results are expressed in µg/kg.

### 2.4. Statistical Analysis of Analytical Data

To compare the results of the experiment and test the differences between the different products, Tukey’s test was performed with SigmaStat v.2.0 software. To assess whether the differences between the values were statistically significant, one-way analysis of variance (ANOVA; *p* < 0.05) was performed. In cases where significant differences were detected between the results of two samples, a *T*-test was performed (*p* < 0.05).

### 2.5. Dietary Exposure to Acrylamide

To estimate AA dietary exposure in subjects aged six to thirty-six months, a dietary exposure assessment was performed using the following formula as reported by Esposito, 2021 [14]:EDI = (Q × C)/BW(1)
where EDI is the estimated daily intake of AA (µg/kg body weight/day); Q is the individual food daily consumption in subjects of different age groups (6, 12, 18, and 36 months (kg/day); C is the concentration of AA in food (µg/kg); and BW is the individual body weight (kg) obtained from World Health Organization data [15] (Appendix A). The mean value between the two sexes was used because, at this age, weight differences are not significant.

Based on the results of this study, the dietary exposure assessment was performed considering the worst-case scenario given by the consumption of the foods with the highest AA concentration.

The risk characterization was performed using a margin of exposure approach. By comparing ${BMDL}_{10}$ with the estimated daily intake of AA, the *MOE* (margin of exposure) can be defined, which indicates the “health alert level” and is calculated using the following formula [14]:(2)$$MOE={BMDL}_{10}/\mathrm{EDI},$$

The *BMDL*_10_ value considered for neurotoxic risk was 0.43 mg/kg BW/day, while a value of 0.17 mg/kg BW/day, derived from evidence of Harder’s gland cancer in mice, was used to assess for carcinogenic risk [10].

## 3. Results and Discussions

Appendix A provides all the information about the samples analyzed: ingredients, nutritional values, production technology, AA concentration expressed as the mean of two determinations ± the standard deviation, and coefficient of variation. Of all 62 samples, AA was found in 45. However, it was not detected in 5 homogenized fruit products, 1 fruit puree, 2 homogenized vegetable products, 2 homogenized ham products, 1 homogenized fish product, 1 snack, 2 powdered milk, 2 growth milk, and 1 freeze-dried homogenized meat products.

### 3.1. Acrylamide Concentration as a Function of Production and Storage Technology

To assess the impact of production and storage techniques on AA concentrations in baby food, the average AA values in the different categories were calculated and then converted into percentage to identify the most determinant technologies in the formation of AA (Figure 1a). The technologies considered were sterilization (in 36 samples including homogenized baby foods—OFR1–OFR9, OVE1–OVE5, OFO1–OFO2, OCP1–OCP3, OCPR1–OCPR4, OCV1–OCV3, OPE1–OPE4; fruit purees—FRU1–FRU4; and milk and cocoa snacks—ME1–ME2), grinding (in 4 samples including semolina—SE1–SE2 and cream of rice—CR1–CR2), drying (in 9 samples including biscuits—B1–B6 and snacks—SN1–SN3), spray drying (in 4 milk powder—LP1–LP4), ultra-high temperature (UHT) (in 4 milk—LC1–LC4), and freeze-drying (in 5 samples including homogenized meat food—LIO1–LIO4 and vegetable broth—BV1).

The technology with the greatest effect on AA production was drying, through which 61.52% of the total AA found in the samples was produced. This agrees with expectations as it is a process that exposes the food to temperatures above 120 °C capable of activating the Maillard reaction [16]. In addition, dried products such as biscuits and snacks, which are cereal matrices, are rich in reducing sugars and asparagine, both of which are precursors of AA [12]. Then, 16.17% of the total AA was found in milled products such as cream of rice and semolina; however, in this case, the presence of the contaminant was due to the raw material chemical composition. Subsequently, sterilization resulted in 11.98% of the total AA owing to the viscous consistency of homogenized foods and purees. In addition, a closed autoclave treatment at a temperature of 121 °C for about 30 min is required to achieve commercial sterility, rather than HTST treatment on a heat exchanger. As a result, as demonstrated in a study on rodent feed, autoclaving causes an increase in AA formation compared to other microbiological remediation technologies [17]. Just over 5% of AA is due to freeze-drying, which involves a reduced exposure to heat, preserving the nutritional and organoleptic characteristics of a product [18]. Finally, 2.69% and 2.20% of AA production is due to the UHT process and spray-drying, respectively; both HTST technologies involve high temperatures for a few seconds, preventing the thermal damage of food [19].

### 3.2. Acrylamide Concentration as a Function of Ingredients

The variation in AA concentration as a function of raw material was studied, comparing products that had undergone the same treatment to eliminate the interferences given by the production process.

Regarding drying, three samples in the biscuit category (Figure 2a) showed the highest concentration of all foods analyzed in the study (B1, B2, B4), confirming the results of other works [20,21,22]. These samples exceeded the 150 µg/kg AA reference value set by regulations. B5 and B6 were biscuits samples showing the least amount of AA in the category. Comparing nutritional labels (Appendix A), they have the least amount of wheat flour. In fact, B5 consists of six different cereal flours, while B6 is 54% made up of spelt flour. These results are in accordance with the study of Miśkiewicz et al., where biscuits made using wheat flour had the greatest concentration of AA [23]. However, the results contrast with another study where biscuits made using wheat, rice, and corn have lower concentration than the ones made of rye, teff, and oat [24]. These discordances are probably due to the different concentrations of asparagine in the cereal matrices used in the recipes of the various biscuits. In fact, within the same cereal species, the concentration of asparagine and the consequent presence of AA in the final product depend on the variety, growing climatic conditions, and processing technologies [25]. Nguyen et al. reported that at biscuits baking temperatures around 200 °C, glucose increases due to the thermal degradation of sucrose, and consequently fructose also increases due to glucose isomerization. Among the reducing sugars, fructose contributes the most to the formation of AA in biscuits; thus, it is confirmed that the interaction between the production process and ingredients is responsible for the presence of AA in foods [12].

Snacks are also included in the dried product category. SN3 showed the highest AA concentration of 25.88 ± 0.21 µg/kg (Figure 2b). It is made entirely of corn flour—a cereal rich in free asparagine, especially when attacked by pathogens, grown under conditions of water stress and sulfur deficiency and in the presence of nitrogenous fertilizers. In addition, corn products are often treated with production processes conducted at high temperatures, and all these aspects contribute to AA formation. The literature shows that the AA concentration in corn-based snacks varies between 5 and 923 µg/kg, perhaps depending on the different cultivation conditions [26].

To assess the impact of the primary ingredient on AA formation in sterilized foods (homogenized baby food in jars and fruit purées), the average AA value of a given category was divided by the total AA found in the sterilized products and then reported as a percentage (Figure 1b).

The lowest amount of AA was found in fruit- and vegetable-based baby foods, probably because of their lower protein and amino acid content compared with meat-, fish-, and cheese-based baby foods. In fruit-based products, the concentration of AA is probably the lowest because—given their acid pH matrices—their sterilization is carried out at temperatures lower than 100 °C, promoting less accumulation of negative substances such as AA [27]. Among them, plum-based ones showed the highest concentration of AA (Figure 2c). In particular, OFR8, made up of 99.9% of plums, has the highest content of AA compared to OFR7 and OFR9, although they are composed of starchy sources in addition to fruits (Appendix A). The increased contaminant concentration can be explained by the fact that plums are an asparagine-rich food matrix [28]. This result agrees with the EFSA assessment, even if it shows higher AA concentrations than those of our study [10]. Concerning vegetable-based homogenized foods (Figure 2d), OVE1 has the highest AA content of 16.80 ± 0.57 µg/kg. However, comparing its label with the other OVEs, there is no compositional evidence to justify this (Appendix A). It is possible to suppose that the ingredients in OVE1 may have undergone a cooking/blanching process for times and/or temperatures higher than the other samples. In contrast, the lower content of AA in OVE2 and OVE3 may be due to a more innovative sterilization technology, such as sterilizers capable of rotating packages to facilitate the diffusion of heat, reducing the exposure time to high temperatures.

Regarding homogenized cheese products, it has been found that OFO2 has a lower concentration of AA (5.96 ± 0.18 µg/kg) than OFO1 (8.89 ± 0.46 µg/kg); the addition of 400 mg calcium in OFO2’s formulation (Appendix A) may be responsible as calcium ions interact with the carboxyl groups of asparagine, inhibiting the formation of the Schiff base and, consequently, the formation of AA [29].

Concerning homogenized fish-based products (Figure 2e), OPE3, consisting of plaice and potatoes, was found to have the highest amount of contaminant, amounting to 45.52 ± 0.28 µg/kg, exceeding the reference value expressed in the regulation. These results can be explained by the ingredients since OPE3 has the highest potato quantity equal to 20%, (Appendix A), and ingredients certainly responsible for the formation of AA [10].

Concerning homogenized meat foods, Figure 2f shows that OCPR and OCV contain small amounts of contaminant, with average AA values that are equal to 1.15 ± 1.34 µg/kg for OCPR and 9.29 ± 3.47 µg/kg for OCV; they consist mostly of 20–30% meat and starchy source (Appendix A). This agrees with other studies, where values below 50 µg/kg have been found [30]. For homogenized chicken foods, the results were rather unexpected, as they showed an AA concentration much higher than the one recommended by the regulation (40 µg/kg). The AA concentration was 211.84 ± 16.53 µg/kg for OCP1, 154.32 ± 12.71 µg/kg for OCP2, and 194.88 ± 7.40 µg/kg for OCP3. Considering the average value, this is the food category with the highest AA content. No study in the literature shows this, so hypotheses must be formulated. Chicken is a rather fibrous, dry, and stringy meat, and it needs to be cooked for a longer time to facilitate the homogenization process and improve the consistency of the finished product. This could be the cause of the higher AA concentration. Furthermore, as it is more susceptible to enzymatic degradation and microbial attack, it may need longer cooking times and temperatures than other meats. Finally, the chicken meat could be rich in asparagine because of the composition of the feed since the diet affects the body composition. The high concentration could be due to the farming conditions, as the chemical composition of the tissues may be affected by physical activity or stress. OCP2 showed the highest percentage of meat and the lowest concentration of AA compared with other OCPs. Thus, it appears that the higher the protein source is, the lower the contaminant content. This was confirmed by other samples. With 20% fish, OPE2 had the lowest AA content (the other OPEs contained 18%). Conversely, OCV2 contained 20% meat and a higher concentration of AA, while the other OCVs contained 30% meat.

Regarding freeze-dried homogenized foodstuffs, AA was mostly found in LIO3 (21.22 ± 1.49 μg/kg), a turkey-based sample. The assumptions made for homogenized chicken products were also extended to this category. Generally, it follows that poultry meat has a higher asparagine content or needs to be cooked for a longer time and at a higher temperature.

Regarding milk (Figure 2g), in powdered formulations, the average AA value identified was 7.94 ± 6.51 μg/kg, in agreement with other studies, where the average AA values were 3.4 μg/kg [31] and 3.21–9.06 μg/kg [32]. In powdered foods, protein’s role is more significant in AA formation than other factors (sugars, moisture, and pH) [33]; this agrees with our results, as LP1 and LP4 showed the highest concentration of AA in this category, having more protein than the others (Appendix A). As for the growth milk (in liquid form), soy-based LC1 had the highest AA content of 11.43 ± 0.45 μg/kg, and this could be due to its higher protein content (2.5 g/100 g) than the other LCs (Appendix A) along with low sulfur amino acid content, high amounts of asparagine and aspartic acid typical of soybeans [34]. Since it has been found that two milk powder and two growing milk samples have no AA, it can be argued that the difference effects of the production processes (UHT and freeze-drying) on AA formation in milk is not relevant. It can also be concluded that these foods do not significantly affect children’s exposure to AA, as confirmed by the study of Boyaci-Gunduz in 2022 [35].

Additionally, two milk and cocoa-based children’s snacks, ME1 and ME2, were analyzed, resulting in AA contents of 40.95 ± 0.32 µg/kg (over the reference limit) and 20.79 ± 0.82 µg/kg, respectively. These results are partially in line with those reported by EFSA in the “ready-to-eat meal and dessert” category, where the average medium bound (MB) level was 20 µg/kg [10].

Finally, for a comprehensive view of AA concentration in baby food, cereal-based foods such as rice cream (CR) and semolina (SE) were analyzed. The results agree with those of the study of Michalak et al. [36]; however, they are lower than those in the EFSA’s report, where the average MB level of AA in baby food based on cereals to be reconstituted was 125 µg/kg [10]. In our study, it was found that the mean value in CR was 21.85 ± 1.41 µg/kg. SE1 exceeded the reference limit of the regulation, with an AA value of 46.07 ± 0.23 µg/kg, while SE2 had a concentration of 26.82 ± 0.26 µg/kg. This difference can be the result of different asparagine contents in the grains used for the two different semolina products due to the variety used, weather conditions of cultivation, harvest year because dry conditions favor AA, and processing technologies [25].

### 3.3. Assessment of Dietary Exposure to Acrylamide in Infants and Children

Since AA has shown probable carcinogenicity in animal studies, EFSA has not established a tolerable daily intake, as any level of exposure could be potentially dangerous. Data in the literature are reassuring regarding the risk of developing toxic effects in the nervous system [36]; these occur following intakes of 100 mg/kg BW/day, many orders of magnitude higher than the daily dose consumed through the diet. However, there remains concern about the risk of cancer occurrence, for which there is no NOAEL (No Observed Adverse Effect Level); thus, even minimal exposure to AA could potentially cause tumor development [8]. Several studies in the literature report that AA intake in children is higher than that in adults [10,20]. This is due to their lower body weight; for the same amount ingested, the exposure per kilogram of BW is higher in infants than in adults. In addition, although most gastrointestinal functions develop within the first year of life, intestinal motility remains slow, and the small intestine remains incompletely developed; thus, greater absorption of toxic elements is noted in infants compared with adults [37]. Therefore, the intake of AA in subjects from six to thirty-six months was estimated in the worst-case scenario of this study, defined by the consumption of biscuits and homogenized chicken products. The estimated daily intake of AA was calculated according to Equation (1) [14].

The individual daily food consumption (Q) (kg) was considered: for six-month-old children 0.01 kg/day biscuits, one jar of homogenized chicken per day (0.08 kg); for twelve-month-old children 0.015 kg/day biscuits, one jar of homogenized chicken per day; for twenty-four-month-old children 0.025 kg/day biscuits, no homogenized food; for thirty-six-month-old children 0.035 kg/day biscuits, no homogenized food. (Data on biscuits consumption was provided by Label B3, while Humana.it [38] provided information on the homogenized chicken food). Regarding the AA concentration value (C), the average values of the biscuits category equal to 161.35 ± 101.59 µg/kg and the average value of the homogenized chicken food category equal to 187.01 ± 29.56 µg/kg were used. Regarding body weight (BW), the mean value between males and females was used, which was equal to 7.6 kg, 9.25 kg, 11.85 kg, and 14.10 kg in6-, 12-, 24-, and 36-month-old subjects, respectively (Appendix A).

Then, to carry out a risk characterization, it was necessary to calculate the *MOE* as reported in Equation (2). According to the CONTAM Panel, *MOE* values below 100 for neurotoxic effects and below 10.000 for carcinogenic effects are of public health concern. In Figure 3, the *MOE*s for neurotoxic effects are above 100 in all cases and thus not of concern. On the other hand, all *MOE*s for carcinogenic effects are below 10.000. Although the carcinogenic effect of AA in humans has not yet been demonstrated, the intake of these foods is alarming given the probable carcinogenicity. The effects of biscuit consumption become more worrying as the subject’s age increases (Figure 3a), as they consume more of this food. Comparing *MOE*s (from biscuits consumption) with those estimated by EFSA, the values are within the range only for 36-month-old subjects, while for the other age groups, the values are beyond the upper margin [10]. Exposure to AA from consuming homogenized chicken products (Figure 3b) can be assessed in subjects up to one year of age, as they tend to diversify their diets and stop eating such products after this age [38]. Their exclusive consumption allows the EFSA-estimated maximum value for daily dietary exposure in children (1.9 µg/kg BW/day) to be reached; therefore, it is necessary to limit their consumption, since AA intake from other foods ingested during the day compounds the situation.

It is important to remember that not all the foods analyzed have such high levels of AA, but the daily intake was evaluated based on the worst-case scenario to see whether it represents a risk. Based on the consumption of infant biscuits and homogenized chicken foods, the consumer should be alarmed by the significant risk associated with exposure to a probable carcinogen, and this agrees with other studies in which a worrying scenario is outlined [14].

## 4. Conclusions

Infants and children are the groups most exposed to AA through diet as reported by the EFSA CONTAM panel. AA is included in IARC Group 2A as a probable carcinogen, but EU Regulation No. 2017/2158 only provides reference levels, not legal limits, and mitigation measures to reduce AA in foodstuffs. As a result, 62 baby foods belonging to the most significant food categories in the diet of children aged 6–36 months were evaluated to determine whether these reference values were respected. It was found that 14.5% of the products considered exceeded these values, in particular homogenized chicken food, three types of biscuits, a homogenized product with plaice and potatoes, a semolina, and a children’s snack with milk and cocoa. Subsequently, the estimated daily intake of AA was assessed in the worst-case scenario reported in the study, given by the intake of biscuits and homogenized chicken food. Emerged values are associated with *MOE*s that are not concerning for neurotoxic effects but are alarming for the probable carcinogenic effects. Since is not possible to define a safe or tolerable daily intake level of AA, it is necessary to make consumers/parents aware of the risk caused by the dietary intake of this contaminant. For this reason, parents are advised to use a homemade preparation that allows control over cooking time and temperature. However vacuum and steaming processes capable of reducing temperatures and time, respectively, should be preferred both in industrial and homemade cooking to limit the risk of AA formation.

In conclusion, this study can be a useful data source for the EFSA to expand scientific opinion on AA in food. With the aim of protecting consumers, such data could be essential to guide institutions toward imposing a legal limit for the AA concentration in food, with more restrictive values in baby foods.

Nevertheless, this research is not exempt from limitations. It would be interesting to extend it by evaluating the impact on AA formation of mild technologies such as high hydrostatic pressures, vacuum cooking, and ultrasound application.

## Figures and Tables

**Figure 1 foods-13-02690-f001:**
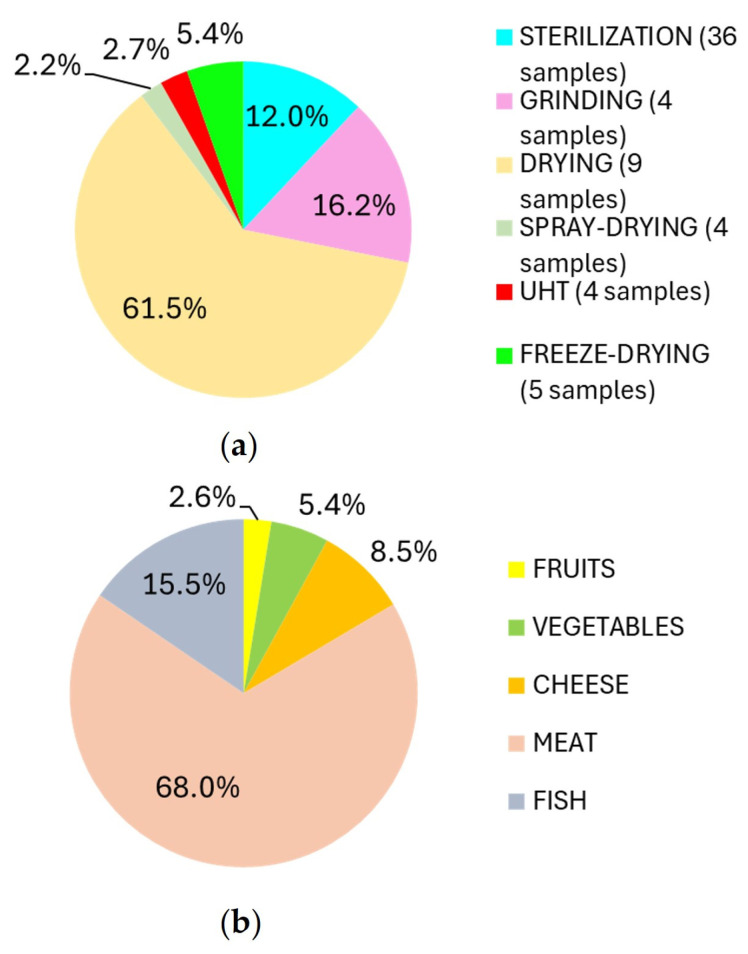
(**a**) The percentages indicate the different influence of production technologies on AA formation. They are the result of the ratio of the average AA value in a production technique category to the sum of the averages, multiplied by 100. Sterilization covers all homogenized foods, fruit purees, and milk-cocoa snacks; grinding includes rice cream and semolina; drying includes biscuits and snacks; spray drying includes powdered milk; UHT includes growth milk; and freeze-drying includes vegetable broth and lyophilized homogenized foods. (**b**) To investigate the influence of raw material on AA formation, products that underwent the same technological process, in this case, sterilization, were compared. The percentages indicate the different influences of the raw material on AA formation in sterilized foodstuffs. The fruits category includes OFR1–OFR9 and FRU1–FRU4; the vegetables category includes OVE1–OVE5; the cheese category includes OFO1–OFO2; the meat category includes OCP1–OCP3, OCPR1–OCPR4 and OCV1–OCV3; and the fish category includes OPE1–OPE4. The percentages are the results of the ratio of the average AA value of each raw material category to the sum of those values, multiplied by 100.

**Figure 2 foods-13-02690-f002:**
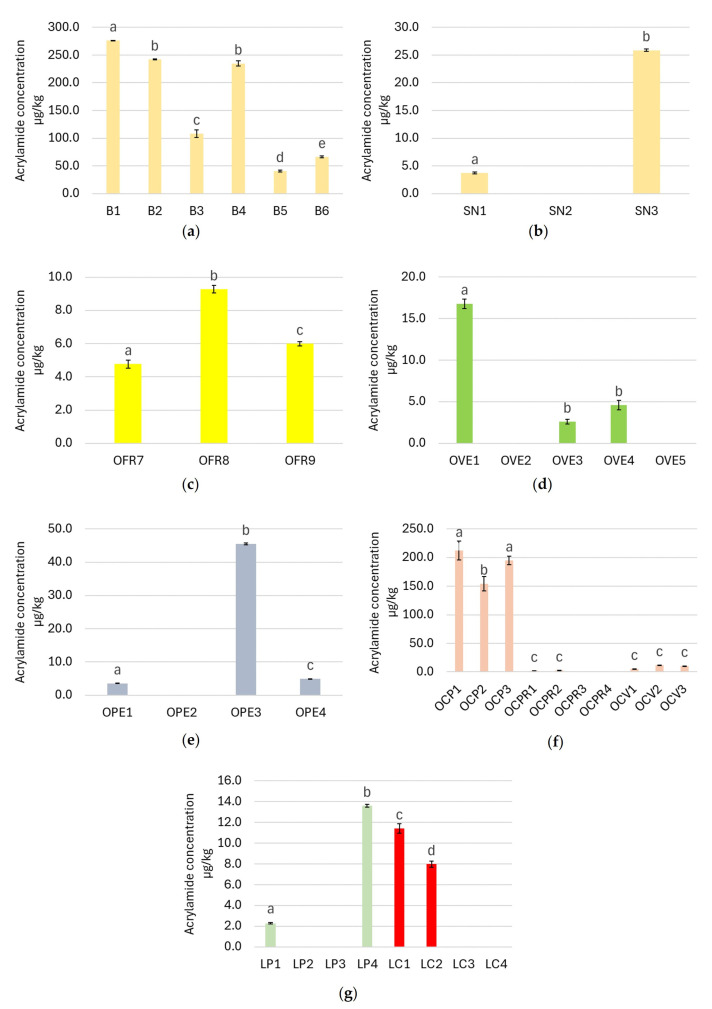
(**a**) AA concentration (µg/kg) in the different samples of biscuits B1–B6 (additional information about their composition can be found in Appendix A). Results are the mean of two determination (technical replicates starting with AA extraction) ± the standard deviation. Different letters (a–e) indicate a statistically significant difference (*p* < 0.05). (**b**) AA concentration (µg/kg) in the different samples of snacks SN1–SN3 (additional information about their composition can be found in Appendix A). The results are the mean of two determinations (technical replicates starting with AA extraction) ± the standard deviation. Different letters (a–b) indicate a statistically significant difference (*p* < 0.05). (**c**) AA concentration (µg/kg) in homogenized plum products OFR7–OFR9 (additional information about their composition can be found in Appendix A). The results are the mean of two determinations (technical replicates starting with AA extraction) ± the standard deviation. Different letters (a–c) indicate a statistically significant difference (*p* < 0.05). (**d**) AA concentration (µg/kg) in homogenized vegetable foods OVE1–OVE5 (additional information about their composition can be found in Appendix A). The results are the mean of two determinations (technical replicates starting with AA extraction) ± the standard deviation. Different letters (a-b) indicate a statistically significant difference (*p* < 0.05). (**e**) AA concentration (µg/kg) in homogenized fish foods OPE1–OPE4 (additional information about their composition can be found in Appendix A). The results are the mean of two determinations (technical replicates starting with AA extraction) ± the standard deviation. Different letters (a–c) indicate a statistically significant difference (*p* < 0.05). (**f**) AA concentrations (µg/kg) in homogenized meat foods OCP1–OCP3, OCPR1–OCPR4, and OCV1–OCV3 (additional information about their composition can be found in Appendix A). The results are the mean of two determinations (technical replicates starting with AA extraction) ± the standard deviation. Different letters (a–c) indicate a statistically significant difference (*p* < 0.05). (**g**) AA concentrations (µg/kg) in powdered milks (LP1–LP4) and liquid milks (LC1–LC4) (additional information about their composition can be found in Appendix A). The results are the mean of two determinations (technical replicates starting with AA extraction) ± the standard deviation. Different letters (a–d) indicate a statistically significant difference (*p* < 0.05).

**Figure 3 foods-13-02690-f003:**
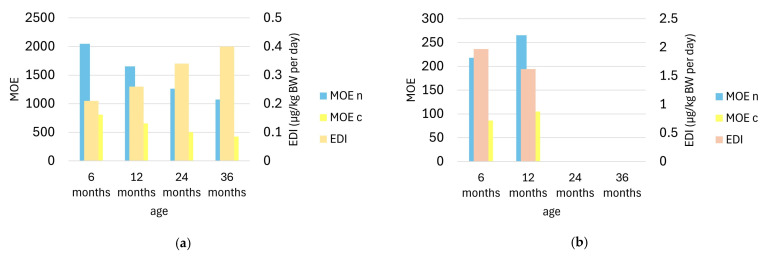
(**a**) Variation in estimated AA daily intake (EDI) (µg/kg BW per day) and *MOE* for neurotoxic (${MOE}_{n}$) and carcinogenic effects (${MOE}_{c}$) based on biscuit consumption as a function of age. (**b**) Variation in EDI (µg/kg BW per day), ${MOE}_{n},\;$and ${MOE}_{c}$ based on consumption of homogenized chicken foods as a function of growth. In both graphs, EDI refers to the right axis, and ${MOE}_{n}$ and ${MOE}_{c}$ refer to the left axis.

## Data Availability

The original contributions presented in the study are included in the article and Appendix A, further inquiries can be directed to the corresponding author.

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
