# Peer review of "Baby Foods: 9 Out of 62 Exceed the Reference Limits for Acrylamide"

_foods, 2024, doi:10.3390/foods13172690_

Round 1

Reviewer 1 Report

Comments and Suggestions for Authors

Very nice paper including a comprehensive introduction, a well described analytical merhods section and a sound discussion and conclusion.

A minor point. The risk of carcinogenic effect of AA can bring parents having the possibility to produce baby food themselves. A small section could be a reference to a guide to avoid the AA production at home for the most risky foods that should be related and publicized by authorities

Reviewer 2 Report

Comments and Suggestions for Authors

Dear authors,

This are my comments and suggestions for a manuscript “Baby Foods: Not All of Them Respect the Reference Limits for Acrylamide” by Bonucci et al.

The study is interesting; it does fit well into the aim and scope of Foods journal. However, this manuscript needs major revision.

The introduction is too long. Materials and methods were poorly described. They should be described with sufficient detail to allow others to replicate and build on published results. Results were not clearly presented. Be specific and precise while inserting changes!

In order to achieve the high quality needed for publication in Foods, I kindly ask you to answer on all my suggestions and comments.

Comments and suggestions:

1.      Title: Rewrite – add numbers (e.g. 10 out of 62) instead of „not all of them“

2.      Abstract: in line 15 „no legal limits“ line 18 „with these values“ and „foods that exceed these values“ Rewrite abstract, if you have set some limits (max. value) or used reference values add these values into the abstract.

3.      Abstract: for food that exceed values insert measured values in the abstract.

4.      Abstract: add in the abstract how many (in numbers or percentage) foods did exceed set limits

5.      Lines 20-24: “dietary exposure to AA was evaluated” how was this conducted? This section of abstract is not clearly presented. In the Materials and methods (M&M) section this was not presented. Insert a paragraph that explains procedure (lines 354-374 should be transferred in the M&M section).

6.      Introduction: line 32 “lower limits” insert some numbers in the main text

7.      Line 35 “reference values” add the numbers in the main text

8.      Lines 39-50: remove this part, it is known how AA is formed

9.      Lines 82-84: insert literature citation that is relevant for this statement

10.   Hypothesis (in the introduction) and answer to hypothesis (in discussion) is missing. Please provide it.

11.   Materials and methods: Section 2.1. insert the number of products per each category

12.   Line 101: “all solid samples were crushed before analysis” – in the main text provide explanation how this was performed

13.   Line 108 -109: In the main text, provide the level of purity for standards acrylamide and methacrylamide, and explain how standard/internal standard solutions were prepared. Were these standards modified/labelled in some way before addition to samples?

14.   Line 115: “shaken” provide RPM and temp

15.   Section 2.3. Did you prepare technical replicates for extraction? Provide this information in the main text. Did you perform LC-MS analysis of several injections per sample? How many? Did you calculate Standard deviation and coefficient of variation? What are LOD and LOQ for the methods? Please provide all this information in the M&M section.

16.   Line 148: in the main text, provide value for all concentration points

17.   Did you use DAD in the method?

18.   Line 154: “significance differences” correct into “significant differences”

19.   Results & discussion: lines 158-164 should be transferred into M&M section

20.   Mention Table S1 in the section 2.1. and before 3.1.

21.   Before 3.1. In the main text, mention the number/percentage of foods with detected AA. Mention in which foods AA was not detected.

22.   In table S1 provide more information: were acrylamide concentrations expressed as average ± stDev or Average ± stError. Provide the N (number of replicates), be precise what type of replicates.

23.   In the table S1 provide additional column about which technologies (lines 161-164) were applied in the production of every product.

24.   From presented text it is not clear how Figure 2 was created, in lines 161-164 insert the number of products for each category and in the legend of fig 2. Insert sample names that were considered for each category

25.   In figure 2 remove the name of graph in the figure “Acrylamide in different technologies”, Figure caption is enough. Figure caption should be rewritten – it has to be more informative.

26.   Apply comment no. 25 to all images.

27.   Figure 3. Explain in figure caption what the categories B1-B6 are and where readers can find additional information on the products.

28.   Figure 5. in the legend of fig 5. Insert sample names that were considered for each category

29.   Insert unit and name on the y-axis in fig 3, 4, 6, 7, 8, 9 (not above the graph, but on the left side of y-axis). Also, apply comment no. 27 to all images. Correct all figure captions “two determinations” – be precise, do you mean 2 analyses on the LC-MS or 2 technical replicates starting with AA extraction? When I see the comments no 24-28: I suggest, creation of one figure with multiple panels in which fig 3, 4, 6, 7, 8 and 9 are merged (the name of panel should be provided in the figure caption, not on the figure). Fig 2 and 5 should also be merged into one image with two panels.

30.   Table 1. Should be transfered in the Supplementary materials as Table S2.

31.   Figure 10. Apply the same corrections as previously mentioned for other images (remove panel name from image and insert units).

32.   At the end of results and discussion section, provide a paragraph that deals with limitations of your study.

Comments on the Quality of English Language

Minor editing of English language required

Reviewer 3 Report

Comments and Suggestions for Authors

1.     The abstract and conclusion of this manuscript lack quantitative data and comparative explanation. It is recommended to strengthen it.

2.     In abstract, mention the IARC classification of acrylamide (AA) as group 2A. and the European Union (EU) Regulation No. 2017/2158. Regarding the characteristics of AA and the control of AA by international organizations, it is recommended to move it to the beginning of the introduction.

3.     Regarding E in E = (Q x C)/BW, and the denominator E in MOE = 𝐵𝑀𝐷𝐿10/E, it is recommended to express it more formally in EDI (estimated daily intake). 

4.     “European Union, EU” has been mentioned at the beginning of this document, just use the abbreviation EU later. It is recommended that all nouns requiring "abbreviation" be carefully checked in the manuscript.

5.    The “Conclusion” of this article contains too much content to focus on. It is recommended to simplify the content and add "research limitations" and "further research plans". It can also explain the significance of the results of this study in public health and food safety.

6.     The format of references is inconsistent with the standard format of MDPI journals. It is recommended to check and adjust.
